# Impact of Intensive Youth Participation in Agriculture on Rural Households' Revenue: Evidence from Rice Farming Households in Nigeria

**Idowu James Fasakin [1], Adebayo Isaiah Ogunniyi [2], Lateef Olalekan Bello [3,4,\*], Djana Mignouna [5], Razack Adeoti [5], Zoumana Bamba [6], Tahirou Abdoulaye [4] and Bola Amoke Awotide [4]**

1. Department of Agricultural Economics, University of Ibadan, Ibadan 200132, Nigeria; ifasakin0271@stu.ui.edu.ng
2. International Fund for Agricultural Development (IFAD), Abuja 90021, Nigeria; a.ogunniyi@ifad.org
3. Department of Global Agricultural Science, The University of Tokyo, Tokyo 113-8657, Japan
4. Social Science and Agribusiness, International Institute of Tropical Agriculture (IITA), Bamako 91094, Mali; t.abdoulaye@cgiar.org (T.A.); b.awotide@cgiar.org (B.A.A.)
5. Social Science and Agribusiness, International Institute of Tropical Agriculture (IITA), Cotonou P.O. Box 08-0932, Benin; d.mignouna@cgiar.org (D.M.); r.adeoti@cgiar.org (R.A.)
6. Country Representative, International Institute of Tropical Agriculture (IITA), Kinshasa 4163, Congo; z.bamba@cgiar.org
\* Correspondence: l.bello@cgiar.org

**Abstract:** The youth unemployment situation is an essential component of the current agricultural policy agenda of the Federal Government of Nigeria. Deep-rooted debates on finding a lasting solution to this problem using agriculture have been targeted as one of the panaceas. Using data from 207 systematically selected rice-producing households, this study employed the Propensity Score Matching method (PSM) and the Inverse Probability Weighted Regression Adjustment method (IPWRA) to examine the effect of intensive youth participation in agriculture on productivity and household revenue in Nigeria. We found that the key factors influencing the decisions of youth to participate in agriculture intensively include the number of years of farming experience, access to credit, membership in social groups, income, and land access. The PSM results indicate that rice productivity could increase by 1088.78 kg/ha if youth decide to intensively participate in agriculture. The IPWRA results show a positive and significant impact of intensive youth participation in agriculture on productivity and revenue. Therefore, our results suggest that efforts by the government and stakeholders towards encouraging flexible accessibility to credit (low-interest and easy repayment) and land without collateral to young people could enhance their participation in intensive agriculture and could subsequently boost productivity and household revenue.

**Keywords:** youth employment; primary occupation; rural households; southern Nigeria

## 1. Introduction

The observed global increase in the youth population and unemployment have become a source of concern and currently attracts considerable attention in many discussions on international development [1]. The National Youth Policy [2] defines youth as Nigerian citizens between 18 and 35 years old. With a national population of about 200 million, Nigeria is the most populated country in Africa and has a high proportion of young people and an increasing rate of youth underemployment and unemployment [3].

Due to limited jobs, youth unemployment continues to be one of the main challenges affecting Nigeria politically, economically, and socially. According to the National Bureau of Statistics (NBS) [4], the youth population (15–35 years of age) in Nigeria is approximately 64 million. More than half (54 percent) of youth are unemployed, with more females being unemployed (52 percent) than males (48 percent). More importantly, many of these youth

are also highly educated, and some are graduates of higher institutions. It is reported that about 1.5 million youth graduate every year [3]. The NBS [4] reported that a substantial proportion of the young people who graduate annually and who are unemployed usually go for jobs that intensify their likelihood of being underemployed. Consequently, to find a lasting solution to this problem, youth unemployment has become a vital component of the recent agricultural policy agenda of the Federal Government of Nigeria. The several ongoing debates about youth unemployment target agriculture as the primary sector to count on to resolve these issues.

Agriculture remains a vital sector in many African countries to promote food security and to alleviate poverty [5–7]. The sector generates approximately 70 percent of rural employment, accounts for over 85 percent of total rural income streams, and contributes to about 25 percent of Nigeria's GDP [8]. Thus, if properly harnessed, agriculture could play a major role in providing sustainable employment and income for the ever-growing youth population in Africa, particularly in Nigeria, where about 69 percent of the youth reside in rural areas and depend on agriculture as their primary means of survival.

Despite the stated importance of agriculture, any policy or program designed to focus on agriculture as a way out of the current youth unemployment situation should be strategic. Public and private sector investments in agriculture need to go beyond the focus on improvements in on-farm productivity to incorporate strategies that will lead to increased revenue and income generation along agricultural value chains. This will enhance employment opportunities and motivate youth to be engaged in agriculture. Youth need to be encouraged to take on agricultural activities with broad economic and market-oriented advantages. For instance, rice production-related activities could be worthy of any unemployed youth's attention. This is largely because rice is one of the most consumed staple food crops in Nigeria [9–11]. In addition to the fact that it is widely cultivated across all of the agroecological zones, it is the only crop with a highly significant increase in consumption, which is mostly due to a shift in consumer preferences, rising income levels, increasing population growth, and rapid urbanization. Therefore, encouraging youth participation in agricultural production, particularly in rice production, could be a source of revenue and a way out of unemployment.

Several studies have assessed youth participation in agriculture in developing countries, including Nigeria. One example is a study conducted by Tiaraiyerari and Krauss [12] on the perceptions of youth involvement in Malaysia's urban agricultural program. The authors found that factors such as optimism about agriculture, career motives, support from family and friends, and perceived barriers significantly and positively influenced youth participation in the program. In contrast, Naamwintome and Bagson [13] found out that young people in Ghana perceived agriculture to be profitable but that they are leaving the agricultural sector due to limited access to essential farm resources such as land and capital.

In Nigeria, Etim and Udo [14] examined the willingness of youth to participate in agricultural activities. The results revealed that age, youth experience in agriculture, household income, and belonging to a social group were positive factors that significantly influenced young people's willingness to participate in agriculture. However, household size negatively influenced participation. In the same vein, Fawole and Ozkan [15] examined the willingness of unemployed graduates in Nigeria to participate in agriculture to solve the problem of youth unemployment. The authors found out that gender, education, having attended agricultural training, and marital status significantly influenced the willingness of youths to participate in agriculture. Many studies (such as [16–18]) examined the determinants of youth participation in agriculture in Nigeria. The results from these studies showed that the positive drivers of youth participation in agricultural activities include age, number of extension visits, membership in a social organization, household size, and farm size. On the other hand, factors such as inadequate access to capital, credit, farm machinery, land, and education hindered youth participation in agriculture. All of these studies were conducted in different locations and have produced mixed results.

In addition to understanding the willingness and determinants of youth participation in agriculture, the impact of youth participation on meaningful livelihood outcomes is more important and should also be a focus of research on this topic. However, literature on the impact of youth participation in agriculture was relatively scarce until a recent study conducted by Bello et al. [19] on the impact of the Youth-In-Agriculture Programme (YIAP) on job creation in Nigeria. However, before pushing for youth to be involved in agriculture, it is important to first evaluate the extent of youth involvement. No study has been conducted to assess this vital aspect of youth participation in agriculture. Thus, there is a gap in the literature that this study intends to fill. Understanding the impact of the level/degree or intensity of youth engagement in agricultural production could also be a serious factor in the decision-making process. The level/degree of their participation could significantly impact the achievable productivity and, by extension, revenue. There could be different revenues between the youth who are involved in agriculture on a full-time (intensive) basis compared to those who are partially (non-intensive) engaged in agriculture as a secondary occupation. Therefore, the main question that this study intends to answer is, are youth participating in agriculture on a full-time basis (intensive) better off than those participating on a part-time (non-intensive) basis? Therefore, this study responds to this question empirically by using the Propensity Score Matching (PSM) and the Inverse Probability Weighted Regression Adjustment (IPWRA) methods while considering the Nigerian context.

The remainder of the paper is organised as follows: Section two presents the materials and methods used in the study, including the analytical framework and estimation techniques. The results and main findings of the study are presented and discussed in section three. Lastly, section four focuses on discussing the conclusions and recommendations.

## 2. Materials and Methods

### 2.1. Study Area

The field survey was conducted in two southern states, Ebonyi (South-East) and Ekiti (Southwest), which were selected from the main rice-producing states in Nigeria. Farming is the principal activity of the inhabitants of these two states. Ebonyi is popularly recognized for its high production of a unique type of local rice known as "Abakaliki", producing an estimated output of 6 tonnes per hectare [20]. Ekiti is also renowned for producing another type of local rice called "Igbemo", with an estimated output of 500 MT/ha [4]. The two states contribute significantly to rice production, processing, and rice value addition in Nigeria [20–22]. The map of the study area is presented in Figure 1.

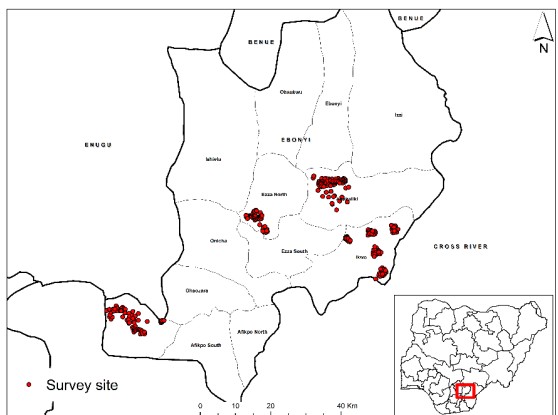 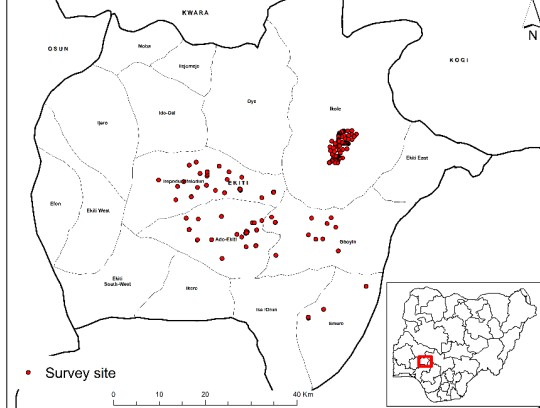

**Figure 1.** Map of Nigeria showing the study areas using the sampled youth GPS coordinates. Source: IITA-GIS units.

### 2.2. Data and Sampling Procedure

Primary data were collected from rural rice-farming households in Ekiti and Ebonyi. The study adopted a multi-stage random sampling technique in order to obtain data from a cross-section of youth who produce rice either as a main or secondary occupation. The first stage was the purposeful selection of Ebonyi and Ekiti from the main rice-producing states in Nigeria. Both states were purposefully selected from two geopolitical zones for the study and are part of the Staple Crop Processing Zones (SCPZ) outlined in the agricultural transformation agenda [21–23]. The second stage involved the purposeful selection of the five foremost rice-producing Local Government Areas (LGAs): five from Ebonyi and five from Ekiti. Abakaliki North Ikwo, Ivo, Afikpo North, and Ezza North were the LGAs that were selected in Ebonyi, while Irepodun/Ifelodun, Gbonyin, Ado Ekiti, Emure, and Ikole were the local government that were selected in Ekiti State. The next stage was the selection of notable rice-producing villages in the selected LGAs. A total of 207 farmers were selected from the two states. The data were obtained through the administration of a well-structured questionnaire. Information related to socioeconomic characteristics, household size, educational level, distance to market, (km) farming experience, farm size, land ownership, extension access, cooperative membership, risk perception, gender, total agriculture income (NGN), total rice output in (kg), total agricultural expenditure (NGN), and transport costs (NGN) was collected from the farmers.

### 2.3. Theoretical Framework and Estimation Strategies

The theoretical framework is based on the agricultural household model, where a youth's decision to be intensively involved in agricultural activities is considered under the general utility maximization framework. Within the framework of a hypothesis of rationality, youth will only participate in agriculture intensively if it increases productivity and adds more to the household's agricultural productivity and income/revenue. Thus, a rational youth is expected to participate in agriculture intensively if the utility derived from such participation is greater than that derived from non-participation. For instance, if the potential outcomes derived from intensive agriculture/rice production is $R_1^*$ and the expected outcomes anticipated from non-intensive participation is $R_0^*$ if we define the change between the anticipated outcomes from intensive participation in agriculture and non-intensive participation as $R_i^*$, then $R_i^* = R_1^* - R_0^*$; therefore, a rational youth would choose to intensively participate in agriculture if $R_i^* > 0$. However, it is impossible to observe $R_i^*$, but it can be specified as a function of observable variables in the following latent mode:

$$R_i^* = P_i\beta + \tau_i, R_i = 1 \text{ if } R_i^* > 0 \tag{1}$$

This implies that youth choose to participate in the employment/agricultural activities that maximize utility, which is subject to their demographic/socioeconomic characteristics and other determinant factors. In order to evaluate the impact of intensive youth participation in agricultural production, we can proceed by assuming that the vector of the outcome variable is a linear function of the explanatory variables $P_i$, and then specify an outcome as follows:

$$R_1 = P_i\alpha + L_i\sigma + \xi_i \tag{2}$$

where $R_1$ denotes a vector of the outcome variable; $P_i$ is a vector of the explanatory variables included in the model; $L$ is the treatment variable (intensive participation) and is equal to 1 if the youth participates intensively in agriculture and 0 otherwise; $\alpha$ and $\sigma$ are the parameters to be estimated; and $\xi_i$ denotes the random error term.

We hypothesized that intensive youth participation in agriculture would create gainful employment for the youth, ensure that they are focused and be devoted to all of the required agricultural improvement activities, enhancing increased productivity and facilitating increased household income. In line with Mathenge et al. [24], the increase in productivity and the resultant increase in household income could be promoted and thus raise households above the poverty threshold. Estimating Equation (1) using the Ordinary

Least Square (OLS) method will yield a bias impact estimate. Noting that tintensive youth participation in agriculture is not randomly distributed, the unobserved characteristics of the youth can simultaneously influence the random error term $\xi_i$ in Equation (1) and the error term $\varepsilon_i$ in Equation (6), thus leading to problems related to the correlation between the two error terms, i.e., corr $(\varepsilon_i, \xi_i) \neq 0$ [25].

In the absence of an experimental setting, we used observational data to estimate the causal effect of the treatment using data on intensive participants (treated) and non-intensive participants (control). As opined by Schreinemachers et al. [26], selection bias is usually a major concern when conducting impact evaluation studies using observational data. In this study, selection bias tends to occur because the choice of participation is, in most cases, not random. The non-randomness of the assignment of youth into the participation status suggests that the average of $R_0$ for intensive participants (L = 1) and $R_0$ for non-intensive participants (L = 0) may differ systematically, even in the absence of intensive participation in agriculture [27–30].

The problem of selection bias is predominantly pertinent for youth participation in agriculture. The youth who participate intensively in agriculture took it upon themselves to become intensive participants. This identifies a possible self-selection bias problem due to the observable and unobservable characteristics of the youth that needs to be addressed to prevent producing biased estimates of howintensive youth participation in agriculture can impact revenue. Many approaches have been adopted in the literature to resolve the issue of selection bias in impact evaluation. These include several matching methods, fixed effects methods for panel data, and instrumental variable regression [31].

We analyzed the impact of theintensive youth participation in agriculture on revenue as measured by the Average Treatment Effect on the Treated (ATET). ATET is a frequently applied approach in the literature for analyzing the counterfactual impact of policies and programs [32–37]. This approach entails estimating the average difference in the outcome variable (revenue), $R$, of the youth who intensively partcipate in agriculture ($L = 1$) and those who do not participate ($L = 0$). This means that the causal effect of the youth participation is equal to the difference between the potential outcome of the treated youth, $R_1$, and that of the non-intensive participants (control youth) $R_0$. However, the expected value of the likely outcome if the youth who are intensively participating in agriculture decided to participate non-intensively cannot be observed directly.

The most commonly adopted approach that excludes the assumptions of distributional form or covariate exogeneity is the Propensity Score Matching (PSM) approach. PSM has been widely adopted in the literature [26,38–42]. This approach assumes that being assigned to intensive participation only depends on the observed covariates and thus tries to eliminate the sample selection bias by putting conditions on these observable covariates. It does so by matching the households with youth who intensively participate in agriculture with one or more non-intensive participants with comparable observable features by summarizing the conditional probability of the intensive participant youth given pre-treatment characteristics.

Afterwards, computing the common support (CS) region is the next step after predicting the propensity score because the ATET and population should only be specified in this region [43]. The CS region is within the maximum and minimum propensity scores of the intensive and non-intensive participants, respectively. It is delineated by cutting off those observations whose propensity scores are higher than the maximum of the control (non-intensive participants) and lower than the minimum of the treated youth (intensive participants) [43]. The implementation of the CS criteria guarantees that any features/characteristics observed in both the treatment and control groups will be similar [44]. This stage is followed by ATET estimation using the matching estimators.

2.3.1. The Propensity Score Matching (PSM) Estimation Techniques

The matching techniques using PSM involve various steps. The basic and most vital step in PSM is propensity score estimation (i.e., the youth's probability of intensive

participation in agriculture) based on similar characteristics using the Logit model. This Logit model is also used to identify the determinants for intensive youth participation in agriculture in the study area. Using the Logit model, the explained variable is binary and is assigned a value of 1 if the youth intensively participates in agriculture and 0 otherwise. The logistic model postulates the probability ($Q_i$) that intensive participation in agriculture is a function of an index

$$Q[(L) = \mathrm{Qr}(L = 1/\mathrm{P}) = \mathrm{E(LP)}] \tag{3}$$

where $L = \{0, 1\}$ denotes the treatment exposure indicator, and P represents the multidimensional vector of the pre-treatment characteristics. These pre-treatment variables were chosen based on literature about including higher-order terms to ensure balanced variables. The dependent variable in the estimated model takes on the following values: $L_i$ = intensive participation in agriculture (intensive participants = 1; non-intensive participants = 0). The explanatory variables are described as follows:

$$L = \beta_0 + \beta_0 P_1 + \beta_0 P_2 + \beta_0 P_3 + \beta_0 P_4 + \beta_0 P_5 + \beta_0 P_6 + \beta_0 P_7 + \beta_0 P_8 + \beta_0 P_9 + \beta_0 P_{10} + \ldots\ldots\ldots \tau_i \tag{4}$$

where $L$ = intensive participation in agriculture (intensive participation = 0, non-intensive participation = 1)

$P_1$ = Household size (numbers);
$P_2$ = Age (years);
$P_3$ = Gender (female = 0 and 0 otherwise);
$P_4$ = Marital status;
P = Years of education (years);
$P_6$ = Farm experience (Years);
$P_7$ = Rice farm experience (Years);
$P_8$ = Pry occupation;
$P_9$ = Farm size (hectare);
$P_{10}$ = Rice ecology;
$P_{11}$ = Access credit;
$P_{12}$ = Association membership;
$P_{13}$ = Access to extension;
$P_{14}$ = Total distance;
$P_{15}$ = Assets owned;
$\tau_i$ = Error term.

### 2.3.2. Propensity Score Matching (PSM)

There are five matching methods used in PSM, namely the Kernel-Based Method (KBM), Nearest Neighbor Method (NNM), radius, stratification, and interval matching. However, the NNM and KBM methods are the most straightforward and most commonly adopted matching methods. In the NNM, the non-treated individuals that are very close to the propensity score of treated individuals are chosen as partners. Kernel matching uses information from all of the non-intensive participant households and constructs the counterfactual outcomes using a weighting function, and reduces variance [43]. After matching, the most prominent evaluation parameter, known as the Average Treatment Effect on the Treated (ATT), which primarily emphasises the effects on those intended treatment observations (intensive participants), is calculated. The ATT is the change between the anticipated outcome values with and without treatment for the youth who are intensively participating in agriculture [43]. It is expressed as

$$\tau_{ATT} = E(\tau/L_i = 1) = E[R_i(1)/L_i = 1] - E[R_i(0)/L_i = 1] \tag{5}$$

where $R_i(1)$ = the potential outcomes when the $i^{th}$ youth intensively participate in agriculture; $R_i(0)$ = the potential outcomes of the $i^{th}$ youth when they do not intensively

participate in agriculture; $L_i$ represents intensive participation; 1= intensive participate and 0 = otherwise. The mean difference between the observable characteristics and control is written as:

$$E[R_i(1)/L_i = 1] - E[R_i(0)/L_i = 0] = \tau_{ATT} + \varepsilon \qquad (6)$$

where $\varepsilon$ is the selection bias

$$\varepsilon = E[R_i(1)/L_i = 1] - E[R_i(0)/L_i = 0] \qquad (7)$$

The true parameter $\tau_{ATT}$ is only identified if the treatment and control outcomes are the same in the absence of intensive participation. It is specified as follows:

$$E[R_i(1)/L_i = 1] - E[R_i(0)/L_i = 0] \qquad (8)$$

After matching, a covariate balancing test is carried out on different matching methods to determine the quality of the PSM. The variables used for the propensity score are only unaffected by intensive participation in agriculture (or expectation of it). The variables are either fixed over time or can be measured before participation in agriculture. These variables are the socioeconomic characteristics of the respondents and other variables. They include the following: intensive participation in agriculture (intensive participation = 1, non-intensive participation = 0), household size (numbers), age (years), gender (female = 0 and 0 otherwise, marital status, years of education, farm experience (years), rice farm experience (years), pry occupation, farm size (hectare), access credit, association membership, access to extension, total distance (metres), and Assets owned.

One noticeable challenge with the application of PSM is that the estimated results will be biased if there is misspecification in the propensity score model [45,46]. To eliminate the above problem in the PSM approach, we adopted the Inverse Probability Weighted Regression Adjustment (IPWRA) method, a variant of PSM. The IPWRA method serves as a credible solution to the potentially biased estimates (ATET) that might originate because of the occurrence of misspecification in the propensity score models [47]. The IPWRA approach allows us to consistently estimate the treatment effect parameters provided we correctly specify only one of the two models (either the outcome or treatment). This model is able to achieve this by combining regression and propensity score methods, and this property is known as a "doubly robust property" [47]. The IPWRA estimators use probability weights to obtain outcome regression parameters that account for the missing data problem, which arises because each youth farmer is only observed in one of the potential outcomes. The adjusted outcome regression parameters are then adopted to calculate the treatment-level means of the predicted outcomes. The contrasts of these means serve as the sources of the estimates of the treatment effects. Following Wooldridge [46], we estimated the propensity score model to generate the propensity score $p\left(s_i, \hat{\delta}\right)$, and then we adopted the regression model, whereby we weighted by the inverse probability. In this study, we adopted a linear outcome function and estimated $(\lambda_1, \phi_1)$ using inverse probability-weighted least squares, which can be expressed as follows:

$$\overset{min}{\lambda_1, \phi_1} \sum_{i=1}^{K}(r_i - \lambda_1 - \phi_1 s_i)/p\left(s_i, \hat{\delta}\right) \text{ If } L_i = 1, \ \lambda_0, \phi_0 \sum_{i=1}^{K}(r_i - \lambda_0 - \phi_0 s_i)/\left(1 - p\left(s_i, \hat{\delta}\right)\right) \text{ If } L_i = 0 \qquad (9)$$

where the propensity score $p\left(s_i, \hat{\delta}\right)$ is the estimated conditional probability of treatment given the youth's observable characteristics; $\lambda_1, \lambda_0, \phi_0,$ and $\phi_1$ are the parameters to be estimated for the intensive and non-intensive participants, and $r_i$ is the potential outcome

variable. The ATET is then estimated by taking the average difference in the predicted values over the treated samples as follows:

$$ATET_{ipwra} = K_L^{-1} \sum_{i=1}^{K_T} \left[ \left( \hat{\lambda}_1 - \hat{\phi}_1 s_i \right) - \left( \hat{\lambda}_0 - \hat{\phi}_0 s_i \right) \right] \tag{10}$$

where $(\hat{\lambda}_1, \hat{\lambda}_0)$ are the estimated inverse probability-weighted parameters for $L = 0$ and $L = 1$, respectively; $K_L$ is the number of treated youth in the sample.

Explicitly, IPWRA is estimated in three distinct stages, leading to improved efficiency. In the first stage, we evaluated the probability of whether a youth will intensively participate in agricultural production using a binary model such as the logit regression model. We then used the predicted probabilities to reweight the sample by the inverse of the likelihood that each youth is in the intensive and non-intensive participation groups. In the second stage, we estimated the expected outcomes for each youth farmer using a weighted outcome model that includes both the observable characteristics utilized in estimating the participation model and additional information. The outcome model was then used to predict the expected outcome for each of the youth farmers twice: once from the perspective (weights) of the probability of intensive participation in agriculture and again from the perspective (weights) of the likelihood of not intensively participating in agriculture. Finally, the mean outcome for those youth farmers who are intensively participating and those who are not intensively participating in agriculture was then calculated, and the observed difference between these two averages was taken as the expected treatment effect. We used STATA 16.0 software to run the models used in this study.

## 3. Results and Discussion

### 3.1. Summary of Statistics

In Table 1, we present the summary statistics of some of the selected variables in the study. Table A1 in the Appendix A explains the variables used in detail.

**Table 1.** Summary statistics of selected variables.

| Variable | Pooled N = 207 | | Intensive Participants: N = 157 | | Non-intensive Participants: N = 50 | | Mean Diff. |
|---|---|---|---|---|---|---|---|
| | Mean | Std. Dev. | Mean | Std. Dev. | Mean | Std. Dev. | |
| Age (years) | 29.82 | 4.40 | 29.78 | 4.41 | 29.94 | 4.41 | −0.16 |
| Gender (1 = male) | 0.81 | – | 0.80 | – | 0.82 | – | −0.02 |
| Married (yes = 1) | 0.56 | – | 0.58 | – | 0.48 | – | 0.1 |
| Number of Years of education | 12.12 | 3.70 | 12.08 | 3.64 | 12.21 | 3.91 | −0.13 |
| Household size | 4.21 | 2.68 | 4.20 | 2.65 | 4.22 | 2.78 | −0.02 ** |
| Farm experience (Years) | 12.08 | 5.69 | 12.50 | 5.99 | 10.78 | 4.44 | 1.72 |
| Formal training (Yes = 1) | 0.25 | – | 0.25 | – | 0.26 | – | −0.01 |
| Total farm size | 4.92 | 4.95 | 4.87 | 4.62 | 5.09 | 5.90 | −0.22 |
| Access to credit | 0.14 | – | 0.09 | – | 0.31 | – | −0.22 |
| Membership of organization | 0.10 | – | 0.09 | – | 0.12 | – | −0.03 |
| Access to extension | 0.05 | – | 0.03 | – | 0.12 | – | −0.09 |
| Awareness of contract farming (Yes = 1) | 0.43 | – | 0.38 | – | 0.58 | – | −0.2 |
| Income from rice production | 716,743.20 | 886,971.90 | 756,385.40 | 874,209.20 | 589,726.50 | 924,419.40 | 166,658.9 |
| Total distance covered to markets | 11.58 | 49.63 | 13.30 | 56.52 | 6.16 | 11.59 | 7.14 |
| Secured land tenure (Yes = 1) | 0.56 | – | 0.53 | – | 0.62 | – | −0.09 |
| Youth in agriculture (Intensively = 1) | 0.76 | 0.43 | | | | | |
| Total revenue (N) | 411,879.50 | 645,268.50 | 469,027.10 | 737,499.60 | 255,089.70 | 192,741.60 | 213,937.4 |
| Yield | 1567.75 | 4220.49 | 1451.75 | 2828.87 | 1932.00 | 7016.39 | −480.25 |

Source: Field survey, 2020. ** denotes significance level at 5%.

We found that in all of the cases considered, the majority (at least 80 percentage points) of the youth were male, indicating that male youth engage in agriculture more

than their female counterparts. This further affirms findings from previous studies [48–50] that male youth are likely to be more flexible towards agriculture and aspire to engage in knowledge-intensive or "modern" agriculture. The study shows that the mean age of the youth was 29 years old, with an average of 12 years of schooling and an average of 4 household members. This result implies that the youth are "older" and have at least a secondary level of education and come from a moderately-sized household. The result is similar for both the intensive and non-intensive participants.

The result show low access to credit facilities among the youth. The pooled results show that 14 percent have access to credit, while just 9 percent of the youth who are intensively involved in agriculture have access to credit versus the 31 percent of non-intensive participants who have access to credit. This result buttresses the argument from previous studies [48,51,52] that poor access to financial facilities is one of the push factors for intensively participating in agriculture. We found extremely poor access to extension and advisory services among the youth. The result shows that less than one-tenth of the youth who are intensively involved in agriculture have access to extension and advisory services. The findings of our study are congruent with those from studies [53,54] examining the linkages between youth in agriculture and access to extension and advisory services.

Other studies [48,50] have noted that access to land and effective tenure security is a strong driver for youth participation in agriculture. This study shows that 56 percent of youth have access to land through inheritance and land purchases. The outcome variables used in this study are the yield and total revenue from rice production. The results show that the average yield from the intensive participants is 1451.75 kg/ha versus 1932.00 kg/ha from the non-intensive participants among the youth rice farmers. Interestingly, the revenue produced by the youth who are intensively involved in agriculture is higher than that of the non-intensive members. We note that a robust inference cannot be deduced from the mean comparison. Hence, the evidence of the impact of participation will be shown later.

### 3.2. The Factors Influencing Intensive Youth Participation in Agricultur—Logistic Regression

This study estimated the factors driving the decision among youth to intensively participate in agriculture using logistic regression (see Table 2). The logit regression model estimates signify an excellent predictor of intensive youth participation in agriculture as shown by the two alternative test results (i.e., the Hosmer and Lemeshow (H-L) and the chi-square) of the goodness of model fit. The H-L goodness of fit test static was 204.96, and it was non-significant ($p$ = 0.248), depicting that the model is a good fit. A standard rule for logit model acceptance is the non-significance of H-L static [55,56]. Furthermore, chi-square static is 33.63 and is statistically significant. Therefore, this implies that all of the predictors included in the model can jointly predict the intensive participation of youth in agriculture. The results show that intensive participation in agriculture is strongly associated with the socioeconomic and demographic characteristics of the youth involved.

We found that years of farming experience, access to credit, membership in social groups, income, and land access are the key factors (positive and negative) that are driving intensive youth participation in agriculture. For instance, years of farming experience were found to be positive and significantly influenced the decision of the youth to intensively participate in agriculture. The marginal effect shows that increasing the years of farming experience increases the likelihood of intensive youth participation in agriculture by 1.3 percent. The probable reason for this result may be associated with the role of consistency in engagement in agricultural-related activities, which can provide substantial credence to the potentials of agriculture and a deeper understanding of the linkages and pathways along agricultural value chains (VCs), especially rice VCs in the case of the youth in this study. These results echoed the findings of Adesina and Eforuoku [16] who determined that years of engaging in farming activities could influence the decision of youth to be intensively involved in agriculture.

**Table 2.** Logistic regression of the factors influencing intensive youth participation in agriculture.

| Variable | Coefficient | Standard Error | Marginal Effect |
|---|---|---|---|
| Age (years) | −0.047 | 0.061 | −0.007 |
| Gender (1 = male) | −0.184 | 0.543 | −0.027 |
| Marital status (1 = married) | 0.006 | 0.527 | 0.001 |
| Years Edu (Years) | 0.057 | 0.054 | 0.009 |
| House size (number) | 0.003 | 0.076 | 0.001 |
| Farm Exp (years) | 0.080 ** | 0.040 | 0.013 ** |
| rcvFormalTraing (1 = yes) | 0.584 | 0.494 | 0.084 |
| accesCredt (1 = yes) | −1.673 *** | 0.505 | 0.345 *** |
| Memshp (1 = yes) | 0.177 * | 0.107 | 0.027 * |
| accesExtn (1 = yes) | −1.291 | 0.821 | −0.268 |
| Income (local currency) | 0.291 *** | 0.112 | 0.045 *** |
| land_acc3 (1 = yes) | 1.353 *** | 0.467 | 0.188 *** |
| Constant | −2.825 | 2.557 | |
| Log-likelihood | −94.719 | | |
| LR chi2 (18) | 33.63 | | |
| Prob > chi2 | 0.000 | | |
| Pseudo R2 | 0.161 | | |

Source: Field survey, 2020. ***, **, and * denote significance levels at 1%, 5%, and 10%.

The results show that access to credit is significant and negatively influences decisions to participate in agriculture intensively. This implies that access to credit decreases the likelihood of youth deciding to engage in agriculture intensively, and the influence is estimated to be reduced by 34.5 percent. Though the results are contrary to a priori expectations, the relationship between credit access and the decision to participate in agriculture is not surprising. The summary statistics show that youth have low access to credit facilities, which could strongly influence their decision and could discourage their aspirations to pursue agriculture (Table 1). The results are consistent with previous assertions [48,51,52] that poor access to credit facilities and/or financial inclusion will reduce the probability of youth engaging in agriculture.

Similarly, years of membership in social groups were positive and significantly influenced the likelihood of youth intensively participating in agriculture. The marginal effect shows that belonging to a social group increases the probability of intensive participation in agriculture by 2.7 percent. A plausible reason for these findings may be associated with the role of social capital in the value of connectedness and trust between youth, which can bring about a positive mindset and attitudinal changes. Having peers in a social group that are involved in agriculture and who are perhaps successful can influence decisions to participate in agriculture, as noted by other studies [17,57].

Youth income had a positive and significant relationship with the decision to participate in agriculture intensively. This implies that an increase in income is likely to increase participation. The marginal effects suggest that there is a probability participation to increase by 4.5 percent if income increases. This is in line with a priori expectations indicating that increased income is a key component driving the process making more flexible when considering a career in agriculture. A possible explanation for these results may be that increased income may assist in the issue of having low access to credit. Hence, it can provide the necessary input and can allow the adoption of improved technologies to enhance productivity. A similar result was obtained in the study by Twumasi et al. [58] and Magagula and Tsvakirai [59].

Limited land accessibility is predominant among youth who engage in agriculture. Young people can access land by purchasing or leasing it or through inheritance. However, leasing land is time-bound (and might be short) and is determined by land tenure practices [48,50]. In this study, we found that access to land has a positive and significant relationship with the decision to participate in agriculture intensively. Our study shows that youth are 18.8 percent more likely to participate in agriculture if they have adequate access to land. These results suggest that land access plays an important role in youth

employment decisions. These results are consistent with previous studies [59–62] that suggested that access to land has a strong influence on youth decisions to be intensively involved in agriculture, especially in production along agricultural value chains.

### 3.3. Impact of Intensive Youth Participation in Agriculture on Productivity and Revenue

In obtaining the PSM estimator through the logit regression, the socioeconomic status of individual youths was used to form matched observational pairs of youth with similar characteristics. Individual youth who were intensively participating in agriculture (the treatment cases) and those who did not intensively participate in agriculture (the controls) were considered. Matching was carried out based on the individual propensity scores for treatment. The propensity score was operationalized as the predicted probability of participation estimated from a logistic regression of intensive youth participation in agriculture based on the predictors. The propensity score is a probability; thus, the mean probability in the treatment for all of the youth was 76.4 percent, i.e., the likelihood that a particular youth will intensively participate in agriculture (treatment assignment) is 76.4 percent for the outcome variable (productivity and revenue) (see Table 3).

**Table 3.** Propensity score for all youths participating in agriculture.

| Variable | Mean | Std. Dev. | Min | Max |
|---|---|---|---|---|
| Propensity score | 0.764 | 0.194 | 0.113 | 0.994 |

Source: Field survey, 2020.

Furthermore, we tested balancing, which is the matching quality, by constructing a common support graph (see Figure 2). The graph shows a considerable overlap in the propensity scores between the treated and control cases, signifying its effectiveness (i.e., the match is good and balanced) [56].

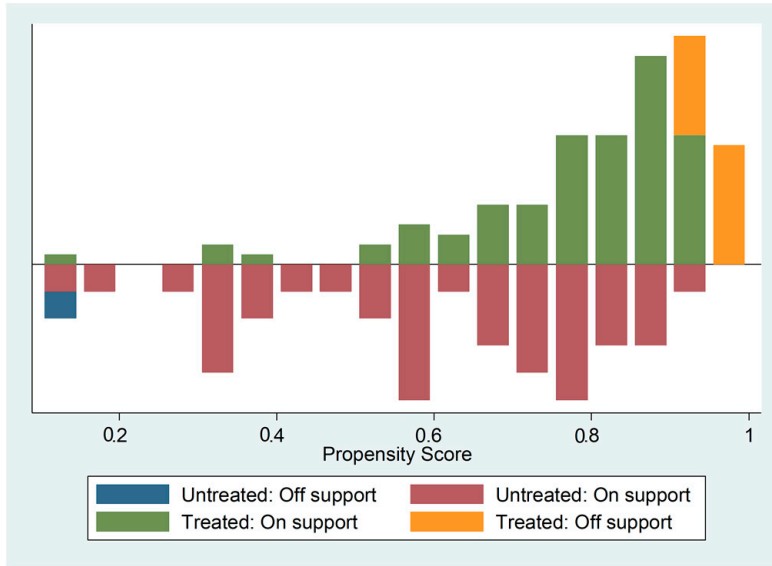

**Figure 2.** Common support graph. Source: Author's computations.

### 3.4. Impact of Youth Participation in Agriculture on Productivity

The impact of intensive youth participation in agriculture on productivity is presented in Table 4. Intensive youth participation in agriculture has a significant effect on rice productivity. According to the Nearest Neighbor matching algorithm, the causal effect of intensive youth participation in agriculture on rice productivity is highly significant and equal to 1088.78, which is the average difference In rice productivity between youth who are intensively engaging in agriculture and non-intensive participants. The results show that the average treatment effect on treated (ATT) is 1088.78 kg/ha. In contrast, if a youth is

chosen at random, then the average treatment effect (ATE) is 841.43, indicating that rice productivity increases by 841.43 kg/ha if any youth is chosen at random to intensively participate in agriculture using the Nearest to Neighbor matching algorithm. The results of the Kernel-based matching method also show a consistent outcome. The results show a positive and highly significant average treatment effect on the treated (ATT). The study shows that the ATT of the youth who are intensively participating in agriculture is 692.18. This implies that intensive participation in agriculture has contributed to and has increased rice productivity of the youth by 692.18 kg/ha.

**Table 4.** Average impact of intensive youth participation in agriculture on productivity.

| Variable Sample | Intensive Participants | Non-Intensive Participants | Difference | S.E. | T-Stat |
|---|---|---|---|---|---|
| *Nearest Neighbor Matching (NNM)* | | | | | |
| Unmatched | 1779.67 | 1137.17 | 642.49 | 282.09 | 2.28 ** |
| ATT | 1779.67 | 690.88 | 1088.78 | 303.37 | 3.59 *** |
| ATU | 1137.18 | 1300.00 | 162.82 | | |
| ATE | | | 841.43 | | |
| *Kernel Based Matching (KBM)* | | | | | |
| Unmatched | 1779.67 | 1137.17 | 642.49 | 282.09 | 2.28 ** |
| ATT | 1572.05 | 879.87 | 692.18 | 296.03 | 2.34 ** |
| ATU | 1167.10 | 1334.79 | 167.69 | | |
| ATE | | | 530.14 | | |

Source: Field survey, 2020. *** and ** denote significance levels at 1% and 5%.

*3.5. Impact of intensive Youth Participation in Agriculture on Crop Revenue*

We also measured the impact of intensive youth participation in agriculture on crop revenue using the Nearest Neighbor and Kernel-based matching algorithm approaches (see Table 5). We found that intensive youth participation can lead to increased rice revenue generation. The Nearest Neighbor and Kernel-based matching algorithms show a positive and highly significant average treatment effect on the treated (ATT). The ATT for intensive youth participation in agriculture increased rice revenue by 249,718.74 in the Nearest to Neighbor algorithm and by 166,028.56 in the Kernel-based matching algorithm. These results imply that intensive youth participation in agriculture can increase rice income by NGN 249,718.74 and NGN 166,028.56, respectively. These results can be justified by considering the impact of youth participation on productivity. Expectedly, a higher yield will lead to more crops being available for sale, possibly increasing crop revenue, especially when there is no market failure. The ATE for the entire population in the sample, that is, when picking any youth at random, was NGN 165,756.99 for the Nearest to Neighbor algorithm and NGN 147,375.05 for the Kernel-based matching algorithm.

**Table 5.** Average impact of intensive youth participation in agriculture on crop revenue.

| Variable Sample | Intensive Participants | Non-Intensive Participants | Difference | S.E. | T-Stat |
|---|---|---|---|---|---|
| **Nearest Neighbor Matching (NNM)** | | | | | |
| Unmatched | 469,027.15 | 255,089.74 | 213,937.41 | 119,794.96 | 2.28 ** |
| ATT | 469,027.15 | 219,308.411 | 249,718.74 | 93,080.55 | 2.68 *** |
| ATU | 255,089.74 | 190,490.128 | 64,599.62 | | |
| ATE | | | 165,756.99 | | |
| **Kernel Based Matching (KBM)** | | | | | |
| Unmatched | 469,027.15 | 255,089.74 | 213,937.41 | 119,794.96 | 1.79 * |
| ATT | 413,181.24 | 247,152.67 | 166,028.56 | 95,331.95 | 1.74 * |
| ATU | 257,855.26 | 363,505.33 | 105,650.07 | | |
| ATE | | | 147,375.05 | | |

Source: Field survey, 2020. ***, **, and * denote significance levels at 1%, 5%, and 10%.

To check and consistency and to validate our PSM findings, we used the Inverse Probability Weighted Regression Adjustment (IPWRA) to estimate the average treatment effect on the treated (ATT) of intensive youth participation on the outcome of interest (productivity and revenue). The results show a positive and highly significant impact of intensive participation in agriculture on both productivity and revenue. Using the log of yield and revenue, the results (see Table 6) show that involvement in agriculture intensively increases productivity and revenue by 3.9 percent and 12.2 percent, respectively. The impact results suggest that intensive engagement in agriculture will play a major role in increasing the productivity of an important staple crop (rice) in Nigeria and will subsequently enhance the household revenue of the country's youth population. Moreover, the COVID-19 pandemic, which has affected different industries, including agriculture, has led to increased unemployment. However, the food industry (agriculture) has remained indispensable for human sustainability. Thus, intensive engagement in agriculture could be an avenue for improving the youth welfare status (income, employment).

**Table 6.** Average impact of intensive youth participation in agriculture on crop revenue—IPWRA.

| Parameters | Productivity | | Revenue | |
| --- | --- | --- | --- | --- |
| | Coefficient | Robust Standard Error | Coefficient | Robust Standard Error |
| ATT | 3.921 *** | 0.456 | 12.200 *** | 0.133 |
| ATE | 0.877 * | 0.527 | 0.211 | 0.182 |

Source: Field survey, 2020. *** and * denote significance levels at 1% and 10%.

## 4. Conclusions

This study empirically examines the impact of the intensive participation of youth in agriculture on crop productivity and rural household revenue. The main findings from the study indicate that years of farming experience, access to credit, years of membership in social groups, income, and land access are the key factors (positive and negative) driving intensive youth participation in agriculture. Furthermore, using a robust econometric technique (PSM and IPWRA), we found a positive and highly significant impact of intensive involvement in agriculture on productivity and revenue among youth farmers.

Therefore, as a policy recommendation, this study suggests the need for relevant organizations (governmental and non-governmental) to provide an enabling environment/policies that will encourage/motivate youth to participate in agriculture more intensively. For instance, access to credit was found to influence youth participation in agriculture; thus, financial institutions (governmental and non-governmental) could revise the existing credit rollout in rural areas. Special credit facilities targeted at youths with low-interest rates and payment flexibility could inspire young people to participate in agriculture intensively. The positive and significant effect of membership in social groups calls for scaling up measures by appropriate agencies (such as agricultural extension services) to form and strengthen youth farmer associations in the study area. Lastly, since land access significantly increases the likelihood of youth participating in agriculture intensively, enacting a policy that will encourage land to be available to young people without collateral should be prioritized by the government and other stakeholders. Such policies should consider non-indigenous young farmers who are not original landowners who intensively participate in agriculture, especially in rice production.

This study is limited to youth farmers engaging in rice production in the southern region of Nigeria. However, the results that emanated from this study showcase the determinants and impact of intensive participation on crop productivity and household revenue in rural areas. Therefore, further research could focus on intensive youth participation in other vital agribusinesses such as other staple and cash crops, animal production, and the processing and marketing of farm produce.

**Author Contributions:** Conceptualization, I.J.F., A.I.O., L.O.B., D.M., R.A., Z.B., T.A. and B.A.A.; methodology, I.J.F., A.I.O., L.O.B. and B.A.A.; software, I.J.F., L.O.B., A.I.O., D.M., R.A., Z.B., T.A. and B.A.A.; validation, I.J.F., A.I.O., L.O.B., D.M., R.A., Z.B., T.A. and B.A.A.; formal analysis, B.A.A.; investigation, I.J.F., A.I.O., L.O.B., D.M., R.A., Z.B., T.A. and B.A.A.; resources, I.J.F., D.M., R.A., Z.B., T.A. and B.A.A.; data curation, I.J.F. and B.A.A.; writing—original draft preparation, I.J.F., A.I.O., L.O.B. and B.A.A.; writing—review and editing, I.J.F., A.I.O., L.O.B., D.M., R.A., Z.B., T.A. and B.A.A.; visualization, I.J.F., A.I.O., L.O.B. and B.A.A.; supervision, D.M., R.A., Z.B., T.A. and B.A.A.; project administration, I.J.F., D.M., R.A., Z.B., T.A. and B.A.A.; funding acquisition, I.J.F. and Z.B. All authors have read and agreed to the published version of the manuscript.

**Funding:** This research was funded by the International Fund for Agricultural Development (IFAD) under the grant 2000001374, and the APC was funded by the International Institute of Tropical Agriculture (IITA).

**Institutional Review Board Statement:** The study was conducted in accordance with the Declaration of Helsinki, and approved by the Institutional Review Board of International Institute of Tropical Agriculture (IRB/IF-CA/001/2021) for studies involving human.

**Informed Consent Statement:** Informed consent was obtained from all subjects involved in the study.

**Acknowledgments:** The authors appreciate the grant awarded by the International Institute for Tropical Agriculture (IITA) and the International Fund for Agricultural Development (IFAD) under the project titled Enhancing Capacity to Apply Research Evidence (CARE) in policy for youth engagement in agribusiness and rural economic activities in Africa. We appreciate Victor Manyong for his technical contribution to this manuscript and also the facilitation of the article APC fee.

**Conflicts of Interest:** The authors declare no conflict of interest.

**Appendix A**

**Table A1.** Definition and description of varibles.

| Variables | Definition of Variables |
| --- | --- |
| Awareness of contract farming | Dummy = 1 if youth is aware of contract farming |
| Contract Farming | Dummy = 1 if youth participates in contract farming |
| Age | Age of the youth rice farmers (years) |
| Gender | Sex of the youth (= 0 if female, 1 = male) |
| Household size | The total household size (number) |
| Source of land | Dummy = 1 if it is personal/owned land |
| Education years | Years of education received by youth farmers (years) |
| Access to credit | Dummy = 1 if the youth has access to credit |
| Access to extension | Dummy = 1 if the farmer has access to extension |
| Membership oforganiztion | Dummy = 1 if the youth is a member of a social organization/group |
| Asset information | Dummy = 1 if the farmer has radio, TV, or mobile phone |
| Primary occupation | Dummy = 1 if the primary occupation is farming |
| Rice production experience | Years of experience in rice farming (years) |
| Farm size | Total area of land cultivated in acres |
| Faring Experience | Years of farming experience |
| Rice output | Output from rice production (kg) |
| Total income from rice | Total income from rice production (Naira, NGN) |
| Income from other crops | Total income from other crops produced (Naira, NGN) |
| Number of people employed | Number of people employed for rice farming production |

Source: Author's computations.

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
