# Peer review of "Impact of Intensive Youth Participation in Agriculture on Rural Households’ Revenue: Evidence from Rice Farming Households in Nigeria"

_agriculture, doi:10.3390/agriculture12050584_

Round 1

Reviewer 1 Report

The manuscript entitled "Impact of intensive youth participation in agriculture on rural households' revenue: Evidence from rice farming households in Nigeria" presents an interesting issue. However, the authors should undertake specific changes that would significantly improve the manuscript and the possibilities of its international dissemination. Among such modifications, I mention the following:

It is recommended to reduce the similarity index. The similarity index after excluding references should preferably be below 15%. Please refer to the attachment for details of the similarity index.

Abstract:

The authors properly presented the background and aim of the study, but they failed to present materials and methods.

Authors should present specific information about the studied population, data gathering, and methodology of analysis.

Authors do not have enough power to formulate recommendations for authorities. Instead, they should formulate conclusions, based directly on the present study and conducted the analysis.

Please add specific numeric data accompanied with p-Values in the abstract.

Please add some conclusions (not only statements reproducing results) at the end of the abstract.

Introduction:

The major problem with the presented Introduction results from the fact that the Authors refer to some references, but not being directly related to the formulated statements. They should change their references into more accurate ones. As an example, I am indicating one of the references (it is only an example and this problem should be corrected within whole section). Authors state that "Agriculture remains the key sector in most African countries for sustainable food security and poverty reduction (Sakketa and Gerber, 2020)" while referring one reference – Rural shadow wages and youth agricultural labor supply in Ethiopia: Evidence from farm panel data. This reference is not directly associated with agriculture in "most African countries". Authors should find more adequate references, presenting this issue, e.g. 
Diao, X., Hazell, P., & Thurlow, J. (2010). The role of agriculture in African development. World development, 38(10), 1375-1383.
Dercon, S., & Gollin, D. (2014). Agriculture in African development: theories and strategies. Annu. Rev. Resour. Econ., 6(1), 471-492.
Clute, R. E. (1982). The role of agriculture in African development. African Studies Review, 25(4), 1-20.
Brzeska, J., Diao, X., Fan, S., & Thurlow, J. (2012). African agriculture and development. Strategies and Priorities for African Agriculture Economywide Perspectives from Country Studies.
The same problem should be corrected in the case of all references.

Results:

Table 1: Authors should clearly define the distribution of the studied variables.

Results should be described clearly – tables present data properly, but they should be properly introduced within the text and commented.

Authors should present in the footnotes all necessary details. In the footnotes to the tables, the authors should point to the original source of the data, not simply the authors' computation.

What part of data was used to create the model and what part of data was used to verified the model. Please specify it.

The font size in Table 3 should be consistent with other tables.

Discussion:
Proper literature-based discussion is necessary.
The current version seems rather to be only draft of the discussion.
Authors may in their discussion include 3 areas: (1) compare gathered data with the results by other authors, (2) formulate implications of the results of their study and studies by other authors, (3) formulate the future areas which should be studied.

Please read the Guidelines for Authors of Agriculture journal carefully, as follows:
"Authors should discuss the results and how they can be interpreted from the perspective of previous studies and of the working hypotheses. The findings and their implications should be discussed in the broadest context possible. Future research directions may also be highlighted."
In addition, the Discussion needs to be a separate section, according to the Guidelines for Authors of Agriculture journal. Therefore, authors need to separate the Discussion from "Results and Discussion" and create a new section.

The format of the references in the text is incorrect, and please correct them to numbers, according to the Guidelines for Authors of Agriculture journal.

Authors should present the limitations of their study (at the end of the Discussions section)

Conclusions:
This section should be brief, and should present major observations from the presented study (2-3 simple sentences based directly on the presented study to present major results and their implications are enough).

Author Response

We appreciate your time in reviewing our manuscript. Please see the attachment 

Reviewer 2 Report

This is a good piece of research which focus upon the youth participation in agriculture and its impact on rural households’ revenue. The research topic has significant contribution in addressing both youth employment and rural development. Below comments should be addressed before acceptation:

  1. Literature review seems to be weak, more international context based literature should be digested to understand the importance of youth employment in rural sector, its social and economic impacts, and the comparison between full-engaged and partly-engaged. Particularly, studies rooted in other developing countries are helpful and should be reviewed. Only focusing on the Nigeria-based literature would narrow research horizon.
  2. Literature review can be embedded in Introduction Section, while can also be set up as a separate Section 2.
  3. Study area: Please add a GIS mapping figure to better present your study area.
  4. It is suggested that Theoretical Framework and Estimation Strategies (Techniques) should be established first, and the Study area and data collection procedures should be presented later.
  5. Policy recommendations can be enriched, and the limitation and deficiencies of this research should be spelled.
  6. For policy recommendation, the literature of “How fire safety management attended during the urbanization process in China?. Journal of Cleaner Production, 236, 117686.” may provide references in terms of proposing policy implications by addressing the regression analysis results.

Author Response

We appreciate your time and efforts in reviewing our manuscript. Please see the attachment 

Reviewer 3 Report

Originality

Research contribution in the paper can be identified; this study can be justified as innovative.

The topic is relevant to the scope of Sustainability.

Title

The title is correct as it reflects the objective of the work. The title is too long.

Abstract

The abstract provides a structured summary including contextual background, and result, conclusion, and implications of key findings, etc.

Keywords

The keywords are adequate.

Introduction

The introduction part does highlight the aim of this investigation. The most relevant part is the introduction section that gives a perfect context for the justification of the research.

Literature review

The main weakness of the paper is that it lacks conceptual clarity, and the literature review should be substantially improved. As a result, the manuscript does not provide a realistic representation of literature that has been published, it does not present sufficient new material to justify publication. The manuscript is insufficiently supported by evidence or proper references of work done elsewhere.

Matherials and Methods

It describes a bit confusingly the methodology, what it was used for and what it was, and how the models come in order. There is no description of the software in it, if I could see with it what it was all about.

Nevertheless, the analysis is good and up-to-date. The results section begins "in medias res" and is cut right in the middle. We know nothing about the sample, its characteristics, and representativeness. It would be good for a power analysis that now this 207 sample is appropriate for this, what is the strength of the model?

How is h1 * different from plain H1? The notation H1 is misleading because hypotheses are usually labeled this way.

I would need a little more about Propensity Score Matching, why this method was used and what it is all about.

Results

Summary, Conclusion, and Policy Recommendations

Please separate the discussion and conclusion parts.

In the conclusion of the paper, it is necessary to connect obtained results with the main purpose of the research and the main hypothesis/research questions.

The Conclusion section should go beyond the interpretation rather than the summary of the results. For instance: - The main objective of this study was to examine .... effects of .... on the........, to shed light on novel research perspectives on ..... - A .................. analysis was used to calculate correlations between ..... while also taking ....... into account ….. - The advantage of the research model is that .......  - Contrary to previous approaches, we take into account -----. We found that ...... . Namely: (a), (b), (c) ....  - The methodological implication is that ........ The findings are also important for policymakers ………. – Limitations - Future research direction.

The manuscript is not well structured, it is not meet the requirements of the Journal (style, citations format, etc.).

Author Response

We appreciate your time  and effort in reviewing our manuscript. Please see the attachment 

Round 2

Reviewer 1 Report

I appreciate the great efforts that the authors have made in response to my questions and concerns. However, there is one issue that should be corrected:
The COVID-19 pandemic has profoundly impacted employment and agricultural production (The pandemic has also impacted youth participation in agriculture), and I recommend that the authors include the following in the Discussion Section: The COVID-19 pandemic has affected different industries, and agriculture is no exception. (please expand on the above)

Author Response

We appreciate your valuable suggestions in improving this manuscripts. We have incorporated your suggestions in line 547 -554.

Reviewer 3 Report

This version including the review comments are fine

 The authors have satisfactorily addressed all the comments.

Author Response

We appreciate your valuable contributions in improving our manuscript.